# Microcirculatory Function during Endotoxemia—A Functional Citrulline-Arginine-NO Pathway and NOS3 Complex Is Essential to Maintain the Microcirculation

**DOI:** 10.3390/ijms222111940

**Published:** 2021-11-03

**Authors:** Karolina A. P. Wijnands, Dennis M. Meesters, Benjamin Vandendriessche, Jacob J. Briedé, Hans M. H. van Eijk, Peter Brouckaert, Anje Cauwels, Wouter H. Lamers, Martijn Poeze

**Affiliations:** 1Department of Surgery, NUTRIM School of Nutrition, Translational Research in Metabolism, Maastricht University Medical Center, 6229 ER Maastricht, The Netherlands; d.meesters@maastrichtuniversity.nl (D.M.M.); hmh.vaneijk@maastrichtuniversity.nl (H.M.H.v.E.); m.poeze@maastrichtuniversity.nl (M.P.); 2Department of Genetics & Cell Biology, NUTRIM School of Nutrition, Translational Research in Metabolism, Maastricht University Medical Center, 6229 ER Maastricht, The Netherlands; 3VIB Inflammation Research Center, 9052 Ghent, Belgium; benjamin.vandendriessche@byteflies.com (B.V.); peter.brouckaert@ugent.be (P.B.); acauwels@orionisbio.com (A.C.); 4Department of Biomedical Molecular Biology, Ghent University, 9000 Ghent, Belgium; 5Byteflies, 2600 Antwerp, Belgium; 6Department of Electrical, Computer and Systems Engineering, Case Western Reserve University, Cleveland, OH 44106, USA; 7Department of Toxicogenomics, GROW School for Oncology and Developmental Biology, Maastricht University Medical Center, 6229 ER Maastricht, The Netherlands; j.briede@maastrichtuniversity.nl; 8Orionis Biosciences, 9052 Ghent, Belgium; 9Department of Anatomy & Embryology, NUTRIM School of Nutrition, Translational Research in Metabolism, Maastricht University Medical Center, 6229 ER Maastricht, The Netherlands; wh.lamers@maastrichtuniversity.nl

**Keywords:** nitric oxide synthase, arginine, microcirculation, endotoxemia, citrulline, NOS3, NOS2

## Abstract

Competition for the amino acid arginine by endothelial nitric-oxide synthase (NOS3) and (pro-)inflammatory NO-synthase (NOS2) during endotoxemia appears essential in the derangement of the microcirculatory flow. This study investigated the role of NOS2 and NOS3 combined with/without citrulline supplementation on the NO-production and microcirculation during endotoxemia. Wildtype (C57BL6/N background; control; *n* = 36), *Nos2*-deficient, (*n* = 40), *Nos3*-deficient (*n* = 39) and *Nos2/Nos3*-deficient mice (*n* = 42) received a continuous intravenous LPS infusion alone (200 μg total, 18 h) or combined with L-citrulline (37.5 mg, last 6 h). The intestinal microcirculatory flow was measured by side-stream dark field (SDF)-imaging. The jejunal intracellular NO production was quantified by in vivo NO-spin trapping combined with electron spin-resonance (ESR) spectrometry. Amino-acid concentrations were measured by high-performance liquid chromatography (HPLC). LPS infusion decreased plasma arginine concentration in control and *Nos3^−/−^* compared to *Nos2^−/−^* mice. Jejunal NO production and the microcirculation were significantly decreased in control and *Nos2^−/−^* mice after LPS infusion. No beneficial effects of L-citrulline supplementation on microcirculatory flow were found in *Nos3^−/−^* or *Nos2^−/−^/Nos3^−/−^* mice. This study confirms that L-citrulline supplementation enhances de novo arginine synthesis and NO production in mice during endotoxemia with a functional NOS3-enzyme (control and *Nos2^−/−^* mice), as this beneficial effect was absent in *Nos3^−/−^* or *Nos2^−/−^/Nos3^−/−^* mice.

## 1. Introduction

Derangement of the microcirculatory flow is one of the critical pathogenic events in sepsis and appears to be an important directly underling cause for the development of multiple organ failure and mortality [1,2,3,4]. These alterations in the microcirculation, if not reversed within the first 24 h after the onset of sepsis (prolonged sepsis), are hypothesized to be the single independent factor in predicting patient outcome [4]. Endothelial damage plays an important role in the development of microcirculatory collapse during prolonged sepsis due to leakage of plasma, and impeded vasodilatation due to a compromised nitric oxide (NO) production [1,2,5,6,7,8,9]. The conversion of arginine to citrulline and NO is the sole source of NO and is mediated by one of the three NO-synthase enzymes (NOS1, NOS2 and NOS3) [10,11,12,13]. Several mechanisms are hypothesized to be responsible for the decrease in endothelial NO production during prolonged endotoxin-induced inflammation. The impaired NO production in the microcirculation is thought to be mediated by dysfunction of NOS3 [14], which is mediated at least partly by arginine deficiency [9,15,16]. This arginine deficiency, in turn, results from an enhanced utilization of arginine upon the pro-inflammatory upregulation of NOS2 and arginase [15,17,18], and from a decreased import of arginine and/or an impaired resynthesis of arginine from citrulline [15,19,20,21,22]. The reduced citrulline bioavailability and production may further deteriorate the already impaired substrate availability in the arginine-NO pathway [15,19,21,22]. Furthermore, excessive NOS2-mediated NO production during endotoxemia in non-homogeneously distributed inflammatory cells results in a maldistribution of NO and a “hyperdynamic” microcirculation by shunting [23].

Supplementation of L-citrulline during endotoxemia restored intracellular arginine availability and improved NO production and microcirculatory function [9]. Based on the intracellular co-localization of NOS3 and ASS [24], It is generally hypothesized, but not yet proven that this beneficial effect of L-citrulline is mediated by endothelial NOS3 [25]. We therefore investigated the effect of L-citrulline supplementation in *Nos2^−/−^*, *Nos3^−/−^*, and *Nos2^−/−^/Nos3^−/−^* mice on intracellular arginine availability, jejunal NO production and microcirculatory flow in a prolonged endotoxemia model.

## 2. Results

### 2.1. Plasma Amino Acid Concentrations in Control and Nos-Deficient Mice under Basal and Endotoxemic Conditions

Under basal conditions *Nos2^−/−^* and *Nos2^−/−^/Nos3^−/−^* mice exhibited significantly lower plasma concentrations of arginine than control and *Nos3^−/−^* mice (Figure 1A). Upon LPS infusion, only the plasma arginine concentrations of control and *Nos3^−/−^* mice decreased significantly (Figure 1A). Plasma citrulline concentrations were similar in all mouse lines under both basal and endotoxemic conditions, except for a higher concentration under basal conditions in *Nos3^−/−^* mice (Figure 1B). Plasma ornithine concentrations were not different between *Nos*-deficient mice and control mice under basal conditions (Figure 1C). LPS administration increased plasma ornithine concentrations in all mouse lines.

L-citrulline supplementation during endotoxemia resulted in enhanced plasma citrulline concentrations in control, *Nos2^−/−^* and *Nos3^−/−^* mice (Figure 1B). Remarkably, the effect was ~20-fold in control, almost twofold less in *Nos2^−/−^* mice, only ~threefold in *Nos3^−/−^* mice, and did not reach significance in *Nos2^−/−^/Nos3^−/−^* mice. Citrulline was effectively metabolized to arginine, as its plasma concentration increased ~fourfold in control + LPS mice and 2-3-fold in *Nos2^−/−^* and *Nos3^−/−^* mice. Even in *Nos2^−/−^/Nos3^−/−^* LPS-Cit supplemented mice, plasma arginine concentrations increased (Figure 1A). L-citrulline supplementation resulted in increased plasma ornithine concentrations in control, *Nos2^−/−^* and *Nos3^−/−^* mice compared to basal and endotoxemic conditions (Figure 1C).

### 2.2. Amino Acid Concentrations in Jejunal Tissue of Control and Nos-Deficient Mice under Basal and Endotoxemic Conditions

Next, we investigated the amino-acid concentrations in jejunal tissue. Under basal conditions, tissue arginine concentrations were similar in *Nos2^−/−^* and *Nos2^−/−^/Nos3^−/−^* mice, but were ~twofold higher in *Nos3^−/−^* mice (Figure 2A). LPS infusion decreased jejunal arginine concentrations in control and *Nos3^−/−^* mice only (*p* < 0.01 and *p* < 0.05, respectively, *n* = 7; Figure 2A). Under basal conditions, jejunal citrulline concentrations were significantly higher in *Nos3^−/−^* mice than in the other mouse lines (Figure 2B). Endotoxemia decreased jejunal citrulline concentration in control and *Nos3^−/−^* mice, whereas it increased jejunal citrulline concentration slightly, but significantly, in *Nos2^−/−^* mice (Figure 2B). Jejunal ornithine concentration was ~1.5-fold higher in *Nos3^−/−^* mice than in control and *Nos2^−/−^/Nos3^−/−^* mice (*p* < 0.05, *n* = 7; Figure 2C). LPS administration increased the intracellular ornithine concentrations in control and *Nos2^−/−^/Nos3^−/−^* mice (Figure 2C).

Citrulline supplementation during endotoxemia resulted in a significant increase in citrulline concentrations in jejunal tissue off all mouse lines (Figure 2B). The increase was ~9-fold for control mice and only ~2-fold for the *Nos*-deficient lines. Reflecting this difference, citrulline supplementation during endotoxemia only enhanced jejunal arginine concentrations in control mice (*p* < 0.01, *n* = 7, Figure 2A). Under the same conditions, jejunal ornithine concentrations were increased in control and *Nos*-deficient mice, but did not reach significance in *Nos2^−/−^/Nos3^−/−^* mice (Figure 2C).

### 2.3. L-Citrulline Enhances the Intestinal NO Production in Control and Nos2-Deficient Mice

Under basal conditions, intestinal intracellular NO concentrations were lower in *Nos*-deficient than in control mice, but reached significance in *Nos2^−/−^**/Nos3^−/−^* mice only (Figure 3). The intestinal NO production in *Nos2^−/−^/Nos3^−/−^* mice was ~twofold lower than that in control mice (*p* < 0.05; Figure 3).

LPS infusion resulted in a significant decrease in the jejunal NO production of control and *Nos2^−/−^* mice (Figure 3, *n* = 5), whereas it increased jejunal NO production ~1.5-fold in *Nos3^−/−^* mice (*p* < 0.05, *n* = 5; Figure 3). Interestingly, the NO production in *Nos2^−/−^/Nos3^−/−^* mice was not altered and remained significantly lower than in control mice. L-citrulline supplementation during endotoxemia enhanced jejunal NO production in control and *Nos2^−/−^* mice (Figure 3, *n* = 5, both *p* < 0.05) and was ~ 3-fold higher in control than in *Nos2^−/−^* mice. The increased NO production due to endotoxemia in *Nos3^−/−^* mice was not further increased by L-citrulline supplementation (*p =* 0.9; Figure 3).

### 2.4. Detrimental Effects of Nos3 Deficiency on Microcirculatory Flow during Endotoxemia

Under basal conditions the total number of perfused vessels and the number of perfused vessels per villus were lower in the *Nos3^−/−^* and *Nos2^−/−^/Nos3^−/−^* mice than in control or *Nos2^−/−^* mice (Figure 4A,B). Endotoxemia reduced the total number of perfused vessels and the number of perfused vessels per villus in control and *Nos2^−/−^* mice, but did not alter these parameters in *Nos3^−/−^* or *Nos2^−/−^/Nos3^−/−^* mice (Figure 4A,B). Under endotoxemic conditions, the total number of perfused vessels and vessels per villus were similar in all mouse lines. L-Citrulline supplementation during endotoxemia increased both parameters in control and *Nos2^−/−^* mice (Figure 4A,B, *p* < 0.05 for both), but not in *Nos3^−/−^* or *Nos2^−/−^/Nos3^−/−^* mice (Figure 4A,B).

## 3. Discussion

The results presented in this study demonstrate that *Nos3* deficiency does not decrease jejunal tissue NO production, but does result in an impaired microcirculation as judged from the decreased number of perfused vessels under basal and endotoxemic conditions. Our findings further reveal that, in the absence of *Nos3*, endotoxemia and supplementation of L-citrulline does not affect the microcirculation, demonstrating that the positive effects of L-citrulline supplementation depend on the presence of a functional NOS3 enzyme. The effects of endotoxemia and L-citrulline supplementation were similar in control and *Nos2^−/−^* mice, which shows that NOS2 does not contribute to the vasodilatory effect of NO in jejunal villi.

The microcirculation is a complex system, in which endothelial NO production plays a key role in regulating blood flow [1,2,4,5,26]. The role of NOS3 in endothelial NO production and microcirculatory homeostasis is well established under physiological conditions, but it has remained unclear thus far whether NOS3-dependent NO production is the exclusive source of NO-dependent microcirculatory flow. Endotoxemic conditions increase *Nos2* expression and decrease that of *Nos3* [27], which is accompanied by the development of an impaired or shunted microvascular perfusion [2,4,5,23,27,28]. Historically, deletion or inhibition of NOS2 was hypothesized to be the key to prevent or treat the detrimental effects of endotoxemia on microvascular perfusion [23,29,30,31]. Normally, endotoxemia results in an increased expression of *Nos2* mRNA and protein, which leads to an increased NO-production [32], but this overexpression of *Nos2* does not enhance hypotension [32,33]. However, due to non-homogeneous expression of *Nos2* in tissues, shunting of the microvascular flow during endotoxemia occurs [23]. A better microvascular responsiveness and a lower mortality during experimental sepsis induced by cecal ligation and puncture were reported in *Nos2*-deficient animals compared to control animals [34]. Our findings are at variance with these data, as the *Nos2*-deficient mice in our study demonstrated an impaired NO-production and a decreased number of perfused vessels during prolonged endotoxemia, similar to control mice. These results indicate that NOS2 does not play a significant role in the intestinal NO-production and the regulation of the microcirculation during prolonged endotoxemia. Furthermore, downregulation of *Nos3* expression during endotoxemia [9] may further contribute to the impaired NO production during endotoxemia in these *Nos2*-deficient animals. In addition, administration of LPS, which induced an increased NOS2-derived NO-production, did not result in an altered microcirculation in *Nos3*-deficient mice. Therefore, our data imply a key function for NOS3 in the maintenance of the intestinal microcirculation under basal and endotoxemic conditions.

The influence of NO on the vascular tone has led to several experimental studies investigating the influence of genetic ablation of *Nos* genes [8,31,35]. In the absence of NOS3, a 50% decrease in basal NO concentrations was observed in brain tissue of *Nos3*-deficient compared to wild-type mice [36]. Despite the lower NO-production in the absence of NOS3, angiogenesis was normal [35]. Compensation by NOS1 was shown to be responsible for the maintenance of an adequate NO production [36,37,38,39,40]. In this respect it is of interest that the observed lower basal NO production in the current study in *Nos3^−/−^* and *Nos2^−/−^/Nos3^−/−^* compared to control mice was accompanied by a decrease in the number of perfused vessels, which indicates that intestinal NOS1 does not compensate for the absence of NOS3, and that NOS3 is essential to maintain the intestinal perfusion. In order to receive a sufficient amount of NOS3-derived NO production, a continuous intracellular arginine availability is key.

This intracellular availability of arginine depends on arginine regeneration from L-citrulline or protein degradation. Regeneration of intracellular arginine from citrulline via argininosuccinate synthetase is essential to maintain NOS3-dependent NO production in endothelial cells [25,41]. As many as three independent intracellular arginine pools have been postulated [24], but more recent literature suggests that, at least in case of enhanced arginine consumption by arginase in endothelial cells, the subcellular location of arginine is less important in maintaining NOS3-derived NO production [42]. We previously observed enhanced arginase activity during endotoxemia [9,43], as determined from the increased ornithine plasma and tissue concentrations in control mice. Furthermore, we showed that, in the absence of arginase-1 in endothelial cells or macrophages, only NOS2-derived NO production significantly increased during endotoxemia, whereas NOS3-derived NO production did not benefit from the arginase-1 deficiency [43]. This discrepancy between our results and those of Elms et al. [42] may result from the enhanced NOS2 activity in arginase-1 deficient animals.

We previously demonstrated that L-citrulline supplementation during endotoxemia resulted in enhanced arginine concentrations both in plasma and tissue [9], while arginine supplementation in this model did not increase tissue arginine concentration. In addition, in *Nos2^−/−^* mice, L-citrulline supplementation exhibited similar effects as in control mice: increased plasma arginine concentrations, an increased intracellular NO production and an enhanced microcirculatory flow. These results suggest that the beneficial effects of L-citrulline supplementation on NO production and microcirculatory flow are mediated by NOS3. The lack of effects of citrulline supplementation in the *Nos2^−/−^/Nos3^−/−^* double knockout mice further implies that NOS1 is not able to compensate for the lack of NOS3. Therefore, our results further underline the subcellular localization of citrulline and NOS3 to maintain a NOS3-specific arginine pool as a substrate for NOS3-derived NO-production in the microcirculation.

To determine the possible potential clinical administration of citrulline and further in vivo implications, we previously conducted a randomized double-blinded control, placebo-controlled crossover study in healthy athletes in which oral citrulline was administrated prior to 1 h of strenuous exercise at 70% of the individual pre-assessed maximal workload capacity. The enhanced arginine availability in the L-citrulline supplemented group resulted in a preserved splanchnic perfusion and reduced intestinal injury during exercise [44]. In line with this previous study, the current study results suggest that L-citrulline is a potential clinical application for patients during endotoxemic or septic conditions accompanied by an impaired microcirculation and arginine deficient state. In conclusion, our study demonstrates that L-citrulline supplementation results in an enhanced de novo arginine synthesis and NO production in control and *Nos2^−/−^* mice during endotoxemia. However, these beneficial effects of L-citrulline supplementation on the microcirculation depend entirely on a functional *Nos3* gene.

## 4. Materials and Methods

### 4.1. Animals

To investigate the role of NOS2 and NOS3 on the microcirculatory flow and the NO production during endotoxemia, male C57Bl6 (control; *n* = 36), *Nos2^−/−^* [45] (*n* = 40), *Nos3^−/−^* [39] (*n* = 39) and *Nos2^−/−^/Nos3^−/−^* mice (*n* = 42) were obtained from the Department for Molecular Biomedical Research, University Ghent, Belgium. The *Nos2^−/−^* [45] and *Nos3^−/−^* mice [39] were initially obtained from Jackson laboratories. *Nos2^−/−^/Nos3^−/−^* double knockout mice were produced at the Department of Biomedical Molecular Biology. All mice were transported to the Centralized Animal Facilities at Maastricht University and allowed a three weeks adaptation period. The mice were two to three months old, weighed 23–28 g and were individually housed at room temperature on a 12 h light-dark cycle, fed standard lab chow (Hope Farms, Woerden, The Netherlands) and water ad libitum until the experimental phase of the study. The protocol was approved by the Committee on the Ethics of Animal Experiments of the Maastricht University Medical Center (Permit Number: 2010-172).

### 4.2. Experimental Design

The experimental protocol and surgical procedures were described in detail elsewhere [9]. In brief, a jugular vein catheter was implanted 4 days prior to the experimental phase of the study. At day four, mice were randomly allocated to receive lipopolysaccharide (LPS; *E. coli* O55:B5, Sigma Aldrich, St. Louis, MO, USA, 0.4 µg × g bodyweight^−1^ × h^−1^; *n* = 109) or sterile 0.9% saline (*n* = 48). In total 13 mice died during the endotoxemia infusion (4 *Nos2^−/−^*, 3 *Nos3^−/−^* and 6 *Nos2^−/−^/Nos3^−/−^* mice). During the final 6 h L-citrulline (6.25 mg/h; *n* = 48; “LPS-Cit” group) or the placebo amino acid L-alanine (12.5 mg/h; *n* = 48; “LPS” group) was administered simultaneously with the LPS infusate. The control group was treated with 0.9% saline and L-alanine (*n* = 48; “control” group). In total, mice received 1.5 mL fluid during the 18 h infusion protocol (83 μL/h). At the end of the 18 h infusion period, mice were anesthetized as described [9]. In vivo tissue NO production (*n* = 60) and jejunal microcirculation (*n* = 84) were measured as described in brief below. Arterial blood was sampled via a cardiac puncture. Organs were then harvested, snap frozen in liquid nitrogen and stored in −80 °C until further analysis.

### 4.3. Plasma and Tissue Amino-Acid Concentration Measurements

After deproteinization, plasma and jejunal amino-acid concentrations were determined by high-performance liquid chromatography (HPLC) [9].

### 4.4. In Vivo Tissue NO Production

The in vivo NO production in jejunal tissue was determined in mice (*n* = 60) injected with spin trap agents and quantified as mono-nitrosyl iron complexes (MNIC) with Fe^2+^-dithiocarbamate complexes with electron spin resonance (ESR) spectroscopy [9]. NO concentrations were calculated from the height of the three-line NO amplitude using Bruker WINEPR software as previously described in more detail [9].

### 4.5. Jejunal Microcirculation Measurements with SDF Imaging

The mucosal microcirculation in the jejunal villi was microscopically visualized with the side-stream dark-field (SDF) imager (Microscan, Amsterdam, the Netherlands) [46,47], as described [9]. In brief, circulation, quantified as the total number of perfused vessels per view and the total number of perfused vessels per villus, was determined using Automated Vascular Analysis software 3.0 (Microscan, Amsterdam, the Netherlands), adjusted according to de Backer et al. [28,48,49]. The average microvascular flow index (MFI), a parameter for the predominant type of flow in the villi in the four quadrants of an image, was scored semiquantitatively (0 = absent, 1 = intermittent, with at least 50% of the time no flow, 2 = sludging, 3 = normal or 4 = hyperdynamic flow) [28]. All imaging experiments were done by an experienced investigator and images were analyzed by two independent blinded experienced researchers.

### 4.6. Statistical Analysis

Statistical analysis of the data was performed using GraphPad Prism 6 (GraphPad, San Diego, CA, USA). One-way analysis of variance (ANOVA) was performed with post-hoc Bonferroni correction between groups to determine significant differences. Data are represented as mean and standard error of the mean (SEM). *p*–values smaller than 0.05 were considered as statistically significant.

## Figures and Tables

**Figure 1 ijms-22-11940-f001:**
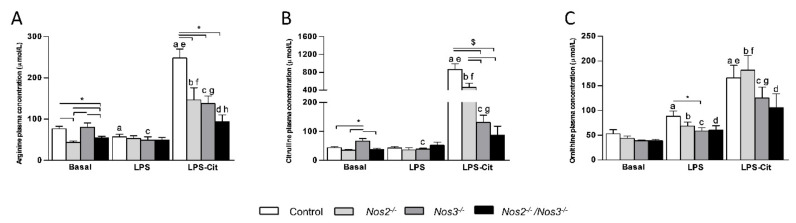
Plasma arginine, citrulline and ornithine concentrations in control and *Nos*-deficient mice under basal and endotoxemic conditions with or without L-citrulline supplementation. (**A**) Plasma arginine concentrations in control and *Nos*-deficient mice under basal and endotoxemic conditions with or without L-citrulline supplementation. (**B**) Plasma citrulline concentrations in control and *Nos*-deficient mice under basal and endotoxemic conditions with or without L-citrulline supplementation. (**C**) Plasma ornithine concentrations in control and *Nos*-deficient mice under basal and endotoxemic conditions with or without L-citrulline supplementation. * *p*-value < 0.05; $ *p*-value < 0.001; a *p*-value < 0.05 vs. control mice under basal conditions; b *p*-value < 0.05 vs. *Nos2^−/−^* mice under basal conditions; c *p*-value < 0.05 vs. *Nos3^−/−^* mice under basal conditions; d *p*-value < 0.05 vs. *Nos2^−/−^/Nos3^−/−^* mice under basal conditions; e *p*-value < 0.05 vs. control + LPS; f *p*-value < 0.05 vs. *Nos2^−/−^* mice + LPS; g *p*-value < 0.05 vs. *Nos3^−/−^* mice + LPS; h *p*-value < 0.05 vs. *Nos2^−/−^/Nos3^−/−^* mice + LPS. Data are shown as mean ± SEM. Statistical significance was determined with one-way ANOVA and post-hoc Bonferroni correction between groups.

**Figure 2 ijms-22-11940-f002:**
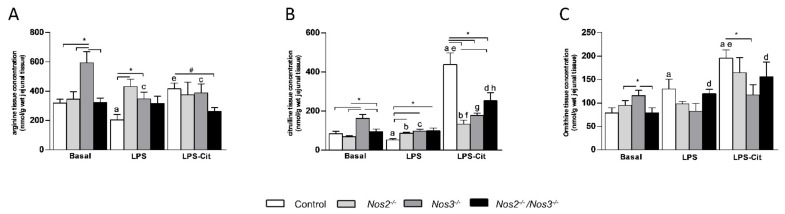
Jejunal arginine, citrulline and ornithine concentrations in control and *Nos*-deficient mice under basal and endotoxemic conditions with or without L-citrulline supplementation. (**A**) Jejunal arginine concentrations in control and *Nos*-deficient mice under basal and endotoxemic conditions with or without L-citrulline supplementation. (**B**) Jejunal citrulline concentrations in control and *Nos*-deficient mice under basal and endotoxemic conditions with or without L-citrulline supplementation. (**C**) Jejunal ornithine concentrations in control and *Nos*-deficient mice under basal and endotoxemic conditions with or without L-citrulline supplementation.* *p*-value < 0.05; ^#^
*p*-value < 0.01; a *p*-value < 0.05 vs. control mice under basal conditions; b *p*-value < 0.05 vs. *Nos2^−/−^* mice under basal conditions; c *p*-value < 0.05 vs. *Nos3^−/−^* mice under basal conditions; d *p*-value < 0.05 vs. *Nos2^−/−^/Nos3^−/−^* mice under basal conditions; e *p*-value < 0.05 vs. control + LPS; f *p*-value < 0.05 vs. *Nos2^−/−^* mice + LPS; g *p*-value < 0.05 vs. *Nos3^−/−^* mice + LPS; h *p*-value < 0.05 vs. *Nos2^−/−^/Nos3^−/−^* mice + LPS. Data are shown as mean ± SEM. Statistical significance was determined with one-way ANOVA and post-hoc Bonferroni correction between groups.

**Figure 3 ijms-22-11940-f003:**
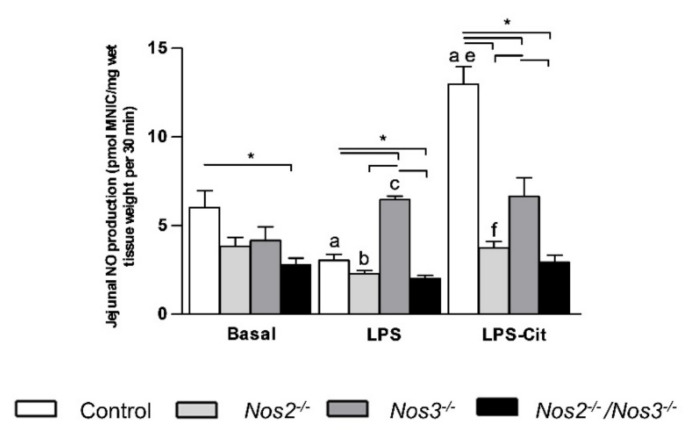
NO production in jejunal tissue of control and *Nos*-deficient mice during basal and endotoxemic conditions, and after L-citrulline supplementation during endotoxemia. Basal jejunal NO production (in pmol MNIC/mg wet jejunal tissue) in control and *Nos*-deficient mice under basal and endotoxemic conditions with or without L-citrulline supplementation. * *p*-value < 0.05; a *p*-value < 0.05 vs. control mice under basal conditions; b *p*-value < 0.05 vs. *Nos2^−/−^* mice under basal conditions; c *p*-value < 0.05 vs. *Nos3^−/−^* mice under basal conditions; e *p*-value < 0.05 vs. control + LPS; f *p*-value < 0.05 vs. *Nos2^−/−^* mice + LPS. Data are shown as mean ± SEM. Statistical significance was determined with one-way ANOVA and post-hoc Bonferroni correction between groups.

**Figure 4 ijms-22-11940-f004:**
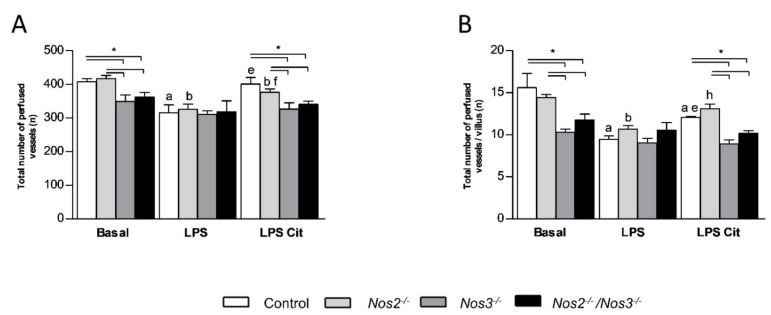
Microcirculatory flow measurements with SDF-imaging in the jejunal villi. (**A**) The total number of perfused vessels in control and *Nos*-deficient mice under basal and endotoxemic conditions with or without L-citrulline supplementation. (**B**) The number of perfused vessels per villus in control and *Nos*-deficient mice under basal and endotoxemic conditions with or without L-citrulline supplementation.* *p*-value < 0.05; a *p*-value < 0.05 vs. control mice during basal conditions; b *p*-value < 0.05 vs. *Nos2^−/−^* mice during basal conditions; e *p*-value < 0.05 vs. control + LPS; f *p*-value < 0.05 vs. *Nos2^−/−^* mice + LPS; h *p*-value < 0.05 vs. *Nos2^−/−^/Nos3^−/−^* mice + LPS. Data are shown as mean ± SEM. Statistical significance was determined with one-way ANOVA and post-hoc Bonferroni correction between groups.

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
