# Peer review of "Microcirculatory Function during Endotoxemia—A Functional Citrulline-Arginine-NO Pathway and NOS3 Complex Is Essential to Maintain the Microcirculation"

_ijms, 2021, doi:10.3390/ijms222111940_

Round 1

Reviewer 1 Report

In the manuscript entitled “Microcirculatory function during endotoxemia—A functional citrulline-arginine-NO pathway and NOS3 complex is essential to maintain the microcirculation” the Authors demonstrated the beneficial effects of L-citrulline supplementation on the microcirculation enhancing the arginine synthesis and NO production in endotoxemia-mice models.

I think that this work is adequately developed and that the results have been rightly demonstrated.

The research in literature does not show a recent paper very similar to this one. In my opinion, this manuscript is favourably considered for publication with few modifications suggested below.

In the Introduction some non-recent bibliographic references are sometimes reported. I suggest to cite more reviews wholly related to what is reported in the text. If it were possible the Authors should replace these references.

Statistical analysis is valuable and performed accurately.

In the Discussion section the Authors should extensively discuss about the insights of pathophysiology, pathologic mechanisms and clinical relevance pointing out the potential clinical administration of citrulline and it’s in vivo implications.

The legends should be shorter without the explanation of  the results.

Lines 198-201: regard on this sentence “in the absence of Nos3, endotoxemia and supplementation of L-citrulline does not affect the microcirculation, demonstrating that the positive effects of L-citrulline supplementation depend on the presence of a functional NOS3 enzyme” I think that it should be useful to analyse the protein levels of NOS3 by western blot.

Author Response

Reviewer 1 comments and suggestions for Authors

The research in literature does not show a recent paper very similar to this one. In my opinion, this manuscript is favorably considered for publication with few modifications suggested below.

  1. In the Introduction some non-recent bibliographic references are sometimes reported. I suggest to cite more reviews wholly related to what is reported in the text. If it were possible the Authors should replace these references.

Answer to comment 1 of reviewer 1:

As requested by the reviewer, where possible, some older references addressing the pathophysiology of sepsis, microcirculation, and organ failure were replaced with newer reviews related to the text.

The changed references are highlighted in the reference list.

  1. In the Discussion section the Authors should extensively discuss about the insights of pathophysiology, pathologic mechanisms and clinical relevance pointing out the potential clinical administration of citrulline and it’s in vivo implications.

Answer to comment 2 of reviewer 1:

As suggested by the reviewer, we added some additional text in the discussion addressing the clinical relevance pointing out the potential clinical administration of citrulline and it’s in vivo implications.

  1. The legends should be shorter without the explanation of the results.

Answer to comment 3 of reviewer 1:

As suggested by the reviewer, we shortened our figure legends and removed the explanation of the results in these legends.

  1. Lines 198-201: regard on this sentence “in the absence of Nos3, endotoxemia and supplementation of L-citrulline does not affect the microcirculation, demonstrating that the positive effects of L-citrulline supplementation depend on the presence of a functional NOS3 enzyme” I think that it should be useful to analyse the protein levels of NOS3 by western blot.

Answer to comment 4 of reviewer 1:

We agree with reviewer 1 that analysis of protein levels of NOS3 by western blot would be interesting. Unfortunately, this is a limitation of our study, as we do not have any tissue samples left to measure the protein levels of NOS3 in this study.

Reviewer 2 Report

Wijnands et al (2021) Microcirculatory function during endotoxemia—A functional citrulline-arginine-NO pathway and NOS3 complex is essential to maintain the microcirculation

The manuscript by Wijnands et al explores the effects of citrulline supplementation on microcirculatory function during endotoxemia and the pivotal role of NOS3. Derangement of microcirculatory flow contributes significantly to sepsis, and subsequent multiple organ failure and mortality. Endothelial NO production by NOSs plays an important role in maintaining microcirculatory flow. Although the role of NOS3 in microcirculatory homeostasis is well-established, it remains unclear whether NOS3 is the exclusive source of NO-production to maintain microcirculatory functions. Authors exploited NOS2-/- and NOS3-/- knock-out and NOS2-/-/NOS3-/- double knock-out mice to investigate the role of NOS2 and NOS3 combined with/without citrulline supplementation in maintaining microcirculation flow during endotoxemia.

They demonstrate that citrulline supplementation results in an enhanced de novo arginine synthesis, intracellular NO-production and microcirculatory flow during endotoxemia in the wild-type and NOS2-deficient mice. However, such beneficial effects are not observed in the NOS3-deficient and NOS2/NOS3-deficient mice.

These results clearly indicate that the beneficial effects of citrulline supplementation on microcirculation during endotoxemia are exclusively dependent on NOS3 function. It also indicates that NO-production by NOS1 cannot compensate for NOS3 as these effects are not observed in the NOS2-/-/NOS3-/- double knock-out mice.

Taken together, this is an excellent study that uncovers the unique role of NOS3 in maintaining microcirculatory flow during endotoxemia upon citrulline supplementation and as such, this reviewer recommends its publication in the International Journal of Molecular Sciences in the present form.

Author Response

Dear Reviewer 2, 

Thank you very much for reviewing our paper and the kind words!